# Effects of School-Based Interventions on Reducing Sugar-Sweetened Beverage Consumption among Chinese Children and Adolescents

**DOI:** 10.3390/nu13061862

**Published:** 2021-05-30

**Authors:** Zhenni Zhu, Chunyan Luo, Shuangxiao Qu, Xiaohui Wei, Jingyuan Feng, Shuo Zhang, Yinyi Wang, Jin Su

**Affiliations:** 1Division of Health Risk Factors Monitoring and Control, Shanghai Municipal Center for Disease Control and Prevention, 1380 West Zhongshan Road, Shanghai 20036, China; zhuzhenni@scdc.sh.cn; 2National Institute for Nutrition and Health, Chinese Center for Disease Control and Prevention, 27 Nan Wei Road, Beijing 100050, China; 3Division of Child and Youth Health, Shanghai Municipal Center for Disease Control and Prevention, 1380 West Zhongshan Road, Shanghai 20036, China; luochunyan@scdc.sh.cn (C.L.); qushuangxiao@scdc.sh.cn (S.Q.); 4School of Public Health, Fudan University, 130 Dongan Road, Shanghai 20030, China; xhwei16@fudan.edu.cn (X.W.); jyfeng16@fudan.edu.cn (J.F.); 5Department of Nutrition, School of Medicine, Shanghai Jiao Tong University, 227 Chongqing South Road, Shanghai 200025, China; zoisite@sjtu.edu.cn; 6Department of Nutrition and Food Science, Steinhardt School of Culture, Education, and Human Development, New York University, 82 Washington Square E, New York, NY 10003, USA; yw5004@nyu.edu

**Keywords:** sugar-sweetened beverage, school-based, intervention, ecological model, difference in difference approach

## Abstract

We set up a series of school-based interventions on the basis of an ecological model targeting sugar-sweetened beverage (SSB) reduction in Chinese elementary and middle schools and evaluated the effects. A total of 1046 students from Chinese elementary and middle schools were randomly recruited in an intervention group, as were 1156 counterparts in a control group. The interventions were conducted in the intervention schools for one year. The participants were orally instructed to answer all the questionnaires by themselves at baseline and after intervention. The difference in difference statistical approach was used to identify the effects exclusively attributable to the interventions. There were differences in grade composition and no difference in sex distribution between the intervention and control groups. After adjusting for age, sex, and group differences at baseline, a significant reduction in SSB intake was found in the intervention group post intervention, with a decrease of 35.0 mL/day (*p* = 0.034). Additionally, the frequency of SSB consumption decreased by 0.2 times/day (*p* = 0.071). The students in the elementary schools with interventions significantly reduced their SSB intake by 61.6 mL/day (*p* = 0.002) and their frequency of SSB consumption by 0.3 times/day (*p* = 0.017) after the intervention. The boys in the intervention group had an intervention effect of a 50.2 mL/day reduction in their SSB intake (*p* = 0.036). School-based interventions were effective in reducing SSB consumption, especially among younger ones. The boys were more responsive to the interventions than the girls. (ChiCTR, ChiCTR1900020781.)

## 1. Introduction

Each year, sugar-sweetened beverage (SSB) intake is related to the loss of about 8.5 million disability-adjusted life years (DALYs), and three quarters of this burden occurs in low- and middle-income countries [1]. SSBs were defined as nonalcoholic beverages sweetened by sugar, such as carbonated beverages, sugar-sweetened fruit or vegetable juice beverages, etc. The frequent consumption of excess amounts of SSBs is a risk factor for obesity [2], type 2 diabetes [3], cardiovascular disease [4], and dental caries [5]. The potential pathophysiology of SSB consumption explained the unfavorable health outcomes linked to SSBs. SSBs contain energy but no micro- and macronutrients, except glucose and fructose. Excessive glucose intake is known for causing adverse glycemic effects on metabolism [6], and fructose can be converted into fat, unfettered by the cellular controls that prevent unrestrained lipid synthesis from glucose [7]. The adverse health outcomes linked to SSBs occur in both adults and children [8,9]. Behavioral factors such as diet linked to mortality causes are preventable and both shape and are shaped by the social environment [10]. Studies have shown that interventions can achieve favorable results in reducing SSB intake in children but not in adults [11]. It is possible that interventions might have effects on individuals who were before or at their behavioral development period rather than those already forming behavioral models.

Comprehensive school-based program achieved some changes on dietary intakes of students [12,13]. School-based interventions aiming at decreasing SSB consumption have shown promising results in previous Western studies [14]. More than half of the children and adolescents in China consumed SSB, and the increasing consumption of SSBs has created the obvious threat of obesity among Chinese children and adolescents [15,16]. However, there has been no study reporting evaluations of school-based interventions tackling SSB reduction in China. Considering the different environmental and cultural backgrounds compared to Western countries, we set up a series of school-based interventions on the basis of an ecological model targeting SSB reduction in Chinese elementary and middle schools and evaluated the effects on SSB reduction among Chinese children and adolescents.

## 2. Materials and Methods

### 2.1. Study Population

We adopted epidemiological methods to establish a parallel control group intervention experiment for one year, from February 2019 to January 2020, in Shanghai, China. Three elementary schools and two middle schools were randomly recruited into the intervention group, and the same number of adjacent counterparts into the control group. Students in the 3rd and 4th grades from the selected elementary schools and the 6th and 7th grades from the selected middle schools were enrolled into the intervention or control group according to their school’s allocation. The participants were not informed of which group they were assigned.

The theoretical minimum sample size was 1051 for the intervention and control groups, respectively, following the formula of sample size calculation for experiments with comparisons between two different groups:n = [(Z_α/2_ + Z_β_)S/δ]^2^

We set α = 0.05 and β = 0.10. S represented standard deviation of SSB intake, andδ represented allowance error.

We decided to select 220 students in each grade of a school and a total of 1100 students in both the intervention and control groups. After enrollment, 1082 and 1259 students were in the intervention and control groups, respectively. By eliminating the students lost to follow up, there were 1046 in the intervention group and 1156 in the control group (Figure 1).

This study was approved by the Shanghai Municipal Center for Disease Control on 17 May 2017 (No. 2017-18). Informed consent was obtained from each participant or the participant’s parents or guardian before the research. The study complied with the code of ethics of the World Medical Association (Declaration of Helsinki). This intervention study was registered at the Chinese Clinical Trial Registry (http://www.chictr.org.cn, accessed on 7 April 2021) with the registration number of ChiCTR1900020781 (http://www.chictr.org.cn/showproj.aspx?proj=35002, accessed on 19 January 2019).

### 2.2. SSB Intake, Frequency, and Knowledge Assessment

The investigators were public health doctors in local community health centers and received a standard training course on the facilitation of the questionnaires. The participants were orally instructed to answer all the questionnaires by themselves. The investigators verified the answers after each questionnaire was completed.

The information collected at the baseline and post one-year intervention were both at the interval between January to February, just before and after the complete one-year intervention period.

A general questionnaire was used to obtain demographic information and SSB-related knowledge. There were six questions related to SSB consumption and health which were used to assess the SSB-related knowledge of the participants. The participant received one point if they gave the right answer. A full score was six. The six questions were as follows:I am not going to get fat if I consume SSBs frequently.SSBs contain many nutrients.SSBs are the best choice for quenching my thirst.I should not have even one sip of an SSB.SSBs bring the body extra energy because the sugar in it can be easily absorbed but they hardly make one feel full.SSBs have empty energy that will not pose a threat to health.

An SSB frequency questionnaire was administrated to collect information on SSB intake and frequency in the past 3 months. This questionnaire originated from a former food frequency questionnaire that was developed and validated by the China Center for Disease and Control and Prevention [17]. SSBs were defined as nonalcoholic beverages sweetened by sugar, excluding fresh juice, categorized as carbonated beverages, sugar-sweetened fruit or vegetable juice beverages, protein-based beverages, probiotic beverages, milk-based beverages, bottled tea beverages, coffee drink, and typical Southeast Asian milky tea (a popular SSB in China). According to each SSB category, SSB consumption frequency was listed in terms of consumption times per day, week, month, or year, and the amount consumed each time was recorded.

The survey was conducted in the pre- and post-intervention periods using the same questionnaires. No disastrous events such as rain or snow disasters that would have affected the normal food supply took place during the research period. The weather was typical of the marine climate in Shanghai, China.

### 2.3. Interventions

The interventions were designed using the structures of an ecological model targeting individuals and their environment, including multiple levels of influence [18]. The interventions were conducted for one year by the teachers in the intervention schools with the collaboration of public health doctors in local community health centers. The control schools had no intervening events. All the participating schools followed the city-wide education schedule during the intervention period.

Individual level■Hold “restricting SSBs”-themed class meeting every semester (twice a year) and praise students with outstanding health behaviors (the themed class meeting was designed to provide SSB-related knowledge regarding the six questions about SSB-related knowledge in the questionnaire).■Distribute SSB-related knowledge materials every semester (twice a year).■Record one’s own SSB behaviors once per week.Family level■Establish WeChat (the most popular social communication application in China) groups for parents and publish new media promotional materials once a month.Peer level■Distribute promotional cards to students every semester (twice a year; the cards showed cartoon figures and information regarding the six questions on SSB-related knowledge in the questionnaire, as well as mottos and goals for the new semester).School level■Carry out a blackboard painting activity with the theme of “understanding SSBs” every semester (twice a year).■Forbid selling SSBs on campus during the intervention.Community level■Negotiate with stores near the gates of schools not to sell SSBs to students.

### 2.4. Statistical Analyses

Statistical analyses were conducted using the SAS statistical software (v. 9.4; SAS Institute, Cary, NC, USA). The comparisons were conducted between the participants in the same level of different schools, different groups and pre and post intervention. T tests were applied to determine the differences between the intervention and control groups pre and post intervention. The difference in difference (DID) statistical approach was used to identify the net effects exclusively attributable to the interventions in this study [19]. The DID approach was designed for quasi-experimental research studying causal relationships in public health settings where randomized controlled trials (RCTs) are infeasible or unethical. In the current analysis of DID, an interactive item of the variate representing group classification and a variate representing pre- or post-intervention classification was included in the multivariate general linear regression models, representing the intervention effect. Age, sex, and the variates mentioned above were included in the same models as covariates. A two-sided *p* < 0.05 was considered to indicate statistical significance.

## 3. Results

### 3.1. Characteristics of the Participants

At baseline, 1082 subjects were recruited in the intervention group and 1259 subjects in the control group. At one-year follow-up, 1046 remained in the intervention group and 1156 in the control group. The retention rates were 96.7% and 91.8% in the intervention and control groups, respectively. Differences in sex distribution were not observed between the two groups either pre or post intervention. The differences in grade composition were significant between the groups both pre and post intervention (Table 1).

### 3.2. Differences in SSB Intake, Frequency, and Knowledge between the Groups pre and Post Intervention

At baseline, the SSB intakes were 286.0 ± 266.5 mL/day and 286.0 ± 288.2 mL/day in the intervention and control groups, respectively. The frequencies of SSB consumption were 1.6 ± 1.6 times/day and 1.7 ± 1.9 times/day, respectively. The scores for SSB-related knowledge were 4.3 ± 1.2 and 4.5 ± 1.0, respectively. There was no difference in the SSB intake and frequency of SSB consumption between the two groups and their subgroups (*p* > 0.05). The scores for SSB-related knowledge were significantly different between the two groups (*p* < 0.001).

After the one-year intervention, the SSB intakes were 220.9 ± 262.3 mL/day and 254.4 ± 268.9 mL/day in the intervention and control groups, respectively. The frequencies of SSB consumption were 1.1 ± 1.5 times/day and 1.4 ± 1.7 times/day, respectively. The scores for SSB-related knowledge were 4.4 ± 1.3 and 4.5 ± 1.1, respectively. Significant differences existed in the SSB intake and the frequency of SSB consumption between groups (*p* = 0.003, *p* < 0.001). The scores for SSB-related knowledge showed no statistical difference between the groups (*p* = 0.060) (Table 2).

### 3.3. The Effects on SSB Intake, Frequency, and Knowledge Attributed to the Interventions

After adjusting for age, sex, and group differences at baseline, a significant reduction in SSB intake which was exclusively attributed to the interventions was found in the intervention group post intervention, with a decrease of 35.0 mL/day (*p* = 0.034). Additionally, the frequency of SSB consumption decreased by 0.2 times/day, with a borderline significance (*p* = 0.071), due to the intervention. No statistically significant effect was found in the score for SSB-related knowledge after the intervention (*p* = 0.347).

After intervention, the participants in the elementary schools with the intervention significantly reduced their SSB intake by 61.6 mL/day (*p* = 0.017) and their frequency of SSB consumption by 0.3 time/day (*p* = 0.002), but no changes were observed in the SSB intake and frequency of SSB consumption among those from middle schools with the intervention (*p* = 0.945 and *p* = 0.978, respectively). The boys in the intervention group had an intervention effect of a 50.2 mL/day reduction in SSB intake (*p* = 0.036), while the girls presented no significant intervention effect in SSB intake (*p* = 0.403) (Table 3).

## 4. Discussion

In the current study, the one-year school-based interventions achieved favorable effects in reducing SSB consumption among Chinese children and adolescents. Regarding the feasibility of the intervention implemented, each participant was not assigned randomly into an intervention or control group. Actually, the participants were allocated indiscriminately into interventions or control groups according to their school’s allocation in the study. Moreover, other citywide health promotion events at the time or the underlying natural growth of individuals might have affected their SSB consumption as well. All these factors made it difficult to identify the effects exclusively attributed to the current interventions. The DID approach provided an alternative means for us to study the net effects on the SSB consumption changes which were exclusively attributable to the interventions [19,20]. After adjusting for age, sex, and group differences at baseline, we discovered a substantial scale of SSB reduction attributable to the interventions in this study. The school stood out as one of the most common settings to improve health behaviors [21]. Previous studies indicated that Western school-based interventions were promising to reduce SSB consumption [14]. Our results give more evidence regarding school-based interventions in the setting of an Eastern culture and environment. The current results also supported the theory that the ecological model helped to develop an environment conducive to change, facilitating the adoption of healthy behaviors [22].

We found that the beneficial effects occurred among the younger children rather than the adolescents in the current study. Both the amount and the frequency of SSB consumption reduced after the one-year intervention, while SSB-related knowledge also increased modestly among the younger children. However, there was no obvious change in the amount and the frequency of SSB consumption and SSB-related knowledge among the participants in the middle schools between pre- and post-intervention. These findings coincided with previous studies showing that interventions focusing on SSB control are more effective on younger children than older ones [23,24]. This also suggests that future interventions or policies aiming at reducing SSBs should target younger children in order to achieve more favorable cost-effective outcomes.

We observed that the consumption of SSBs decreased in the boy participants but not in the girl participants. In the current study, the consumption of SSBs was discrepant at baseline in that the boys consumed more SSBs than the girls no matter whether they were in the intervention or control group. Boys had a higher preference for SSBs and consumed more SSBs than girls in their daily lives [15,25,26]. This might explain the significant effects on the boys after the intervention while there were hardly any effects on the girls in our study. Furthermore, the prevalence of overweight and obesity is much higher in boys than in girls in China [27]. The discrepancies between boys and girls in terms of intervention effects should be taken into consideration when conceiving obesity control strategies among children and adolescents.

In this study, we failed to discover enough evidence of improvement in SSB-related knowledge among our participants through the interventions. Only a modest increase in knowledge score that was of borderline significance (*p* = 0.066) occurred among the participants in the elementary schools with the intervention. In the current interventions, we provided SSB-related knowledge through routine school activities to the students and electronic messages to their parents, which aimed to set up supportive environments for SSB restriction. Previous studies showed that knowledge was not associated with SSB behaviors among children [28]. Other factors such as parental health behavior might determine SSB consumption among children [29,30]. Parental modelling was more crucial to children’s behavioral development [31]. These might be the reasons why the behavior of SSB consumption changed but the SSB-related knowledge remained unimproved among the participants after the intervention in our study.

A limitation of this study was the methodology used to assess SSB intake and frequency. We designed an SSB consumption questionnaire to obtain SSB intake and frequency, which were self-reported by the participants, thus the data on SSB intake and frequency were limited by the accuracy of participants’ estimation and recall. Furthermore, although we adjusted for potential confounding factors, we did not treat dietary intake as one of the confounders under the limitation of data collection. This might have caused bias in the current results. Besides this, the investigators were not blinded to the allocation of schools as intervention and control. This awareness of the investigators might have caused data collection bias. Finally, it was possible that some of the six questions used to assess the SSB-related knowledge were beyond the understanding of the students, which might have influenced our assessment of the SSB-related knowledge change.

## 5. Conclusions

School-based interventions designed in an ecological model were effective in the reduction in SSB consumption among Chinese children and adolescents, especially among younger children. The boys were more responsive to the interventions than the girls.

## Figures and Tables

**Figure 1 nutrients-13-01862-f001:**
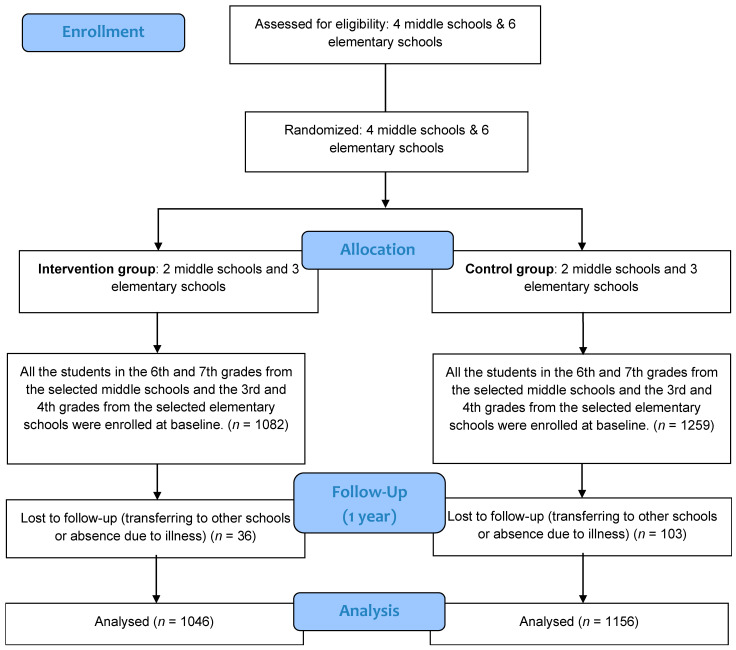
Flow chart of subjects throughout the research.

**Table 1 nutrients-13-01862-t001:** Characteristics of the participants pre and post intervention (%).

	Pre-Intervention			Post-Intervention		
	Intervention Group	Control Group	*p*	Intervention Group	Control Group	*p*
*n*	1082	1259		1046	1156	
Sex						
Boys	51.1	52.0	0.658	51.5	51.6	0.840
Girls	48.9	48.0		48.5	48.4	
Grade at baseline						
3 (9–10 yrs)	31.6	38.8	<0.001	31.6	38.7	<0.001
4 (10–11 yrs)	32.5	30.7		32.5	30.8	
6 (12–13 yrs)	17.7	18.2		17.8	18.2	
7 (13–14 yrs)	18.1	12.3		18.1	12.3	

**Table 2 nutrients-13-01862-t002:** The differences in SSB intake, frequency, and knowledge between the groups pre and post intervention.

		Pre-Intervention			Post-Intervention		
		Intervention Group	Control Group	*p*	Intervention Group	Control Group	*p*
SSB intake, mL/day							
All		286.0 ± 266.5	286.0 ± 288.2	1.000	220.9 ± 262.3	254.4 ± 268.9	0.003
Sex							
	Boys	297.0 ± 272.8	313.6 ± 299.5	0.334	226.2 ± 263.8	301.6 ± 290.3	<0.001
	Girls	274.5 ± 259.5	257.2 ± 273.3	0.292	232.1 ± 263.5	227.9 ± 240.0	0.790
Grade at baseline							
	3 (9–10 yrs)	276.6 ± 280.1	266.4 ± 280.1	0.618	196.2 ± 272.0	229.3 ± 264.9	0.088
	4 (10–11 yrs)	302.1 ± 258.5	277.9 ± 289.2	0.250	189.6 ± 231.9	245.0 ± 258.4	0.003
	6 (12–13 yrs)	256.6 ± 246.3	291.9 ± 279.2	0.185	226.1 ± 231.9	299.0 ± 271.6	0.004
	7 (13–14 yrs)	303.3 ± 274.9	361.1 ± 314.6	0.081	316.1 ± 301.3	315.7 ± 305.2	0.991
Frequency of SSB consumption, times/day							
All		1.6 ± 1.6	1.7 ± 1.9	0.096	1.1 ± 1.5	1.4 ± 1.7	<0.001
Sex							
	Boys	1.6 ± 1.7	1.8 ± 1.9	0.060	1.1 ± 1.4	1.6 ± 1.8	<0.001
	Girls	1.6 ± 1.6	1.6 ± 2.0	0.667	1.2 ± 1.5	1.4 ± 1.7	0.070
Grade at baseline							
	3 (9–10 yrs)	1.7 ± 1.8	1.7 ± 1.9	0.922	1.1 ± 1.6	1.5 ± 1.9	0.005
	4 (10–11 yrs)	1.7 ± 1.6	1.8 ± 2.1	0.501	1.0 ± 1.4	1.4 ± 1.6	0.002
	6 (12–13 yrs)	1.3 ± 1.4	1.7 ± 1.8	0.021	1.1 ± 1.3	1.3 ± 1.4	0.089
	7 (13–14 yrs)	1.5 ± 1.6	1.6 ± 1.6	0.571	1.3 ± 1.5	1.7 ± 1.9	0.127
Score of SSB-related knowledge							
All		4.3 ± 1.2	4.5 ± 1.0	<0.001	4.4 ± 1.3	4.5 ± 1.1	0.060
Sex							
	Boys	4.2 ± 1.2	4.4 ± 1.1	0.006	4.3 ± 1.3	4.4 ± 1.2	0.447
	Girls	4.4 ± 1.2	4.6 ± 1.0	0.006	4.5 ± 1.2	4.6 ± 1.0	0.038
Grade at baseline							
	3 (9–10 yrs)	4.0 ± 1.3	4.3 ± 1.1	<0.001	4.4 ± 1.3	4.4 ± 1.1	0.582
	4 (10–11 yrs)	4.3 ± 1.1	4.5 ± 1.0	0.016	4.2 ± 1.3	4.5 ± 1.1	0.004
	6 (12–13 yrs)	4.7 ± 1.0	4.7 ± 1.0	0.851	4.5 ± 1.2	4.6 ± 1.1	0.421
	7 (13–14 yrs)	4.6 ± 1.0	4.8 ± 0.9	0.137	4.5 ± 1.3	4.7 ± 1.2	0.146

SSB, sugar-sweetened beverage.

**Table 3 nutrients-13-01862-t003:** The effects attributable to the interventions on SSB intake, frequency, and knowledge after the one-year intervention ^a^.

	SSB Intake, mL/day			Frequency of SSB Consumption, times/day		Scores of SSB-Related Knowledge		
	β	*p*		β	*p*	β	*p*	
All	−35.0	0.034		−0.2	0.071	0.1	0.347	
School-level								
Primary school(Grade 3–4, 9–11 yrs)	−61.6	0.002		−0.3	0.017	0.2		0.066
Middle school (Grade 3–4,12–14 yrs)	2.1	0.945		0.0	0.978	−0.1		0.545
Sex								
Boys	−50.2	0.036		−0.2	0.110	0.1		0.262
Girls	−18.8	0.403		−0.1	0.335	0.0		0.888

^a^ β represented the net change of the indicator attributable to the interventions only post intervention compared with that pre-intervention.

## Data Availability

The datasets used and analyzed in the current study are available from the corresponding author on reasonable request.

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
