# Peer review of "Effects of School-Based Interventions on Reducing Sugar-Sweetened Beverage Consumption among Chinese Children and Adolescents"

_nutrients, 2021, doi:10.3390/nu13061862_

Round 1
Reviewer 1 Report
Line 67 - do not end in a preposition. Change "The participants were not informed of which 67 group they were assigned to" to: "The participants were not informed to which 67 group they were assigned"
Please describe how you calculated your sample size. The numbers are fine, but share in your methods what formula you used.
In addition, in the methods, it is difficult to know if you compared the 3rd and 4th grade students to the 6th and 7th grade students. It gets clearer in the figure, but reword to make it clear that you are comparing elementary school children to one another and middle school children to one another.
I am confused, was this a question/statement to the students: "I am not going to get fat if I consume SSBs frequently." It seems out of place in your manuscript. Please explain. Thanks!
Please be sure you include the following references in your manuscript:
A.M. Siega-Riz, L. El ghormli, C. Mobley, B. Gillis, D. Stadler, J. Hartstein, S.L. Volpe, A.L. Virus, J. Bridgman; for the HEALTHY Study Group. The effects of the HEALTHY study intervention on middle school student dietary intakes. International Journal of Behavioral Nutrition and Physical Activity, February 4, 2011 [Epub ahead of print]; Formal publication citation: 8(1):7-15, 2011
The HEALTHY Study Group. A school-based intervention for diabetes risk reduction. The New England Journal of Medicine, June 27, 2010, nejm.org
Author Response
Dear Reviewer:
Thank you for the comments. We had revised the manuscript according to the comments after a serious discussion between the authors. The responses were given as follows:
Line 67 - do not end in a preposition. Change "The participants were not informed of which 67 group they were assigned to" to: "The participants were not informed to which 67 group they were assigned"
Response: Revised as the reviewer suggested.
Please describe how you calculated your sample size. The numbers are fine, but share in your methods what formula you used.
Response: The formula used to calculate the sample size was added into the ‘2.1. Study Population’ part.
In addition, in the methods, it is difficult to know if you compared the 3rd and 4th grade students to the 6th and 7th grade students. It gets clearer in the figure, but reword to make it clear that you are comparing elementary school children to one another and middle school children to one another.
Response: An explanation “The comparisons were conducted between the participants in the same level of different schools, different groups and pre and post intervention.” was added in the methods part to make the meaning clearer.
I am confused, was this a question/statement to the students: "I am not going to get fat if I consume SSBs frequently." It seems out of place in your manuscript. Please explain. Thanks!
Response: The six questions were asked in terms of true or false questions. As for the question "I am not going to get fat if I consume SSBs frequently.", students could answer ‘yes’ for agreement or ‘no’ for disagreement.
Please be sure you include the following references in your manuscript:
A.M. Siega-Riz, L. El ghormli, C. Mobley, B. Gillis, D. Stadler, J. Hartstein, S.L. Volpe, A.L. Virus, J. Bridgman; for the HEALTHY Study Group. The effects of the HEALTHY study intervention on middle school student dietary intakes. International Journal of Behavioral Nutrition and Physical Activity, February 4, 2011 [Epub ahead of print]; Formal publication citation: 8(1):7-15, 2011
The HEALTHY Study Group. A school-based intervention for diabetes risk reduction. The New England Journal of Medicine, June 27, 2010, nejm.org
Response: The two references had been included into the manuscript.

Reviewer 2 Report
I would like to thank the Authors for the well prepared manuscript. The topic is significant and the research was conducted on a large group of participants, which makes that the work is valuable. Due to the obligation of the reviewer, I would like to present my comments that will help in improving the quality of the work.
Abstract
Please remove abbreviations, they should not be used in the abstract. Please add a full list of all abbreviations at the end of the manuscript.
There is no information about the difference between the study group and the control group. What was the main type of methodology- plase clearly state that was the survey method before and after the intervention.
Introduction
At the beginning, please add an SSB explanation - which drinks should be included in this group, give examples and informaton which have the biggest and less amount of sugar.
There are several references to the results of adults, while the work is about children. I recommend to delete information about adults.
There is no short information about the situation of SSB consumption by children in China. Why did the authors decide to investigate this problem.
At the end of the introduction, a clearly defined purpose of the work should appear. Please add the aims of the study.
Methodology
„Study Population chapter” - I miss information about the difference between the study group and the control group. There is no clearly defined criterion for inclusion and exclusion in the study group. Do the Authors take into acount f.ex. children's body weight or comorbidities?
For what reasons were some of the children excluded from the study?
Please add an information about the average time between 1 and 2 tests. Was it exactly 1 year for everyone?
Results
Table 1- is a characteristic of the study group, there is no need to divide into before and after the intervention because these data do not change (age, gender) - do the authors have information about the children's body weight, BMI? it would be worth adding them in this table
Maybe dividing to the information about study and control group would be more appropriate.
Page 6- the source should be corrected (error)
Please improve the visual side of the tables, especially first columns are not readable, please make larger spaces between individual sections
Discussion
line 222- "We found that the beneficial effects occurred among the younger children rather than 222
the adolescents”- please explain why? What is the axplaination of the Authors? give more results of other authors, compare these results with each other, what are the results in China and in the West.
The paragraph (line 239) should be significantly expanded – please write more about the role of parents and school institutions - what are the observations of other authors from China and the West?
Author Response
Dear Reviewer:
Thank you for the comments. We had revised the manuscript according to the comments after a serious discussion between the authors. The responses were given as follows:
Abstract
Please remove abbreviations, they should not be used in the abstract. Please add a full list of all abbreviations at the end of the manuscript.
Response: The abbreviations were removed and a list of all abbreviations was added at the end of the manuscript.
There is no information about the difference between the study group and the control group. What was the main type of methodology- please clearly state that was the survey method before and after the intervention.
Response: The pertinent information was added into the part of Abstract.
Introduction
At the beginning, please add an SSB explanation - which drinks should be included in this group, give examples and informaton which have the biggest and less amount of sugar.
Response: The explanation of SSB was added into the Introduction part as “SSBs were defined as nonalcoholic beverages sweetened by sugar excluding fresh juice, such as carbonated beverages, sugar-sweetened fruit or vegetable juice beverages, etc..”
There are several references to the results of adults, while the work is about children. I recommend to delete information about adults.
Response: We were try to describe the adverse health outcomes linked to SSBs exist in both adults and children, so we retained the references about adults.
There is no short information about the situation of SSB consumption by children in China. Why did the authors decide to investigate this problem.
Response: A short phrase “More than half of the children and adolescents in China consumed SSB” supported by the same reference about a national survey in China into the manuscript (listed blew) was added.
Gui, Z.; Zhu, Y.; Cai, L.; Sun, F.; Ma, Y.; Jing, J.; Chen, Y. Sugar-Sweetened beverage consumption and risks of obesity and hypertension in chinese children and adolescents: A national Cross-Sectional analysis. Nutrients. 2017, 9, 1302, DOI: 10.3390/nu9121302.
At the end of the introduction, a clearly defined purpose of the work should appear. Please add the aims of the study.
Response: A clearer sentence was presented “we set up a series of school-based interventions on the basis of an ecological model targeting SSB reduction in Chinese elementary and middle schools and evaluated the effects on SSB reduction among Chinese children and adolescents.”
Methodology
„Study Population chapter” - I miss information about the difference between the study group and the control group. There is no clearly defined criterion for inclusion and exclusion in the study group. Do the Authors take into acount f.ex. children's body weight or comorbidities?
For what reasons were some of the children excluded from the study?
Response: We enrolled all the students in the 3rd and 4th grades from the selected elementary schools or the 6th and 7th grades from the selected middle schools. There was no special exclusion criteria. The children excluded from the analysis were not present in school during the survey period. The absence reasons were transferring to another school or off school because of illness.
Please add an information about the average time between 1 and 2 tests. Was it exactly 1 year for everyone?
Response: There was only one package of interventions designed using the structures of an ecological model in the current study. The intervention period was including 2 school semester (there are 2 semesters in Chinese school, spring semester and autumn semester) and the vacations.
Results
Table 1- is a characteristic of the study group, there is no need to divide into before and after the intervention because these data do not change (age, gender) - do the authors have information about the children's body weight, BMI? it would be worth adding them in this table
Maybe dividing to the information about study and control group would be more appropriate.
Response: We tried to explain the retention rates were high for there was almost the same situation before and after the intervention. We did not collect the information of body weight.
Page 6- the source should be corrected (error)
Please improve the visual side of the tables, especially first columns are not readable, please make larger spaces between individual sections
Response: In the version of word which we uploaded, we could read all the words in tables. There might be problems in the submission system.
Discussion
line 222- "We found that the beneficial effects occurred among the younger children rather than 222
the adolescents”- please explain why? What is the axplaination of the Authors? give more results of other authors, compare these results with each other, what are the results in China and in the West.
Response: In our study, we found both the amount and the frequency of SSB consumption reduced after the one-year intervention among the younger children but not among the participants in the middle schools. We had added more detailed information in this part. Other studies’ results were cited in the same paragraph.
The paragraph (line 239) should be significantly expanded – please write more about the role of parents and school institutions - what are the observations of other authors from China and the West?
Response: “In the current interventions, we provided SSB-related knowledge through routine school activities to the students and electronic messages to their parents, which aimed to set up supportive environments for SSB restriction.” was added into the paragraph to make it more clear. Other studies’ conclusions were cited as references in the same paragraph.

Reviewer 3 Report
This is a very nice and significant intervention based study for reduction in SSB consumption in School students of 2 different age groups. The study is very nicely presented even though like other population based studies there are some limitations in the study mentioned by the authors themselves. Still author managed to scoop out very good conclusion based on the DID method.

Author Response
Dear Reviewer:
Many thanks for the kind comments.
Round 2
Reviewer 2 Report
I would like to thank the Authors for the manuscript after revision. I accept the manuscript in the present form.